# Reproducibility Study of the Thermoplastic Resin Transfer Molding Process for Glass Fiber Reinforced Polyamide 6 Composites [note 1]

**DOI:** 10.3390/ma16134652

**Published:** 2023-06-28

**Authors:** Filipe P. Martins, Laura Santos, Ricardo Torcato, Paulo S. Lima, José M. Oliveira

**Affiliations:** 1EMaRT Group—Emerging: Materials, Research, Technology, School of Design, Management and Production Technologies Northern Aveiro, University of Aveiro, Estrada do Cercal, 449 Santiago de Riba Ul, 3720-509 Oliveira de Azeméis, Portugal; 2CICECO Aveiro—Institute of Materials, University of Aveiro, Campus Universitário de Santiago, 3810-193 Aveiro, Portugal; 3TEMA—Center for Mechanical Technology and Automation, Mechanical Engineering Department, University of Aveiro, Campus Universitário de Santiago, 3810-193 Aveiro, Portugal

**Keywords:** polyamide 6 composites, thermoplastic matrix, T-RTM, in situ polymerization, glass fiber

## Abstract

Polyamide 6 (PA6) thermoplastic composites have higher recyclability potential when compared to conventional thermoset composites. A disruptive liquid molding manufacturing technology named Thermoplastic Resin Transfer Molding (T-RTM) can be used for processing composites due to the low viscosity of the monomers and additives. In this process, polymerization, crystallization and shrinkage occur almost at the same time. If these phenomena are not controlled, they can compromise the reproducibility and homogeneity of the parts. This work studied the influence of packing pressure, as a process variable, throughout the filling and polymerization stages. To assess the process reproducibility and parts’ homogeneity, physical, thermal and mechanical properties were analyzed in different areas of neat PA6 and composite parts. This study showed that a two-stage packing pressure can be successfully used to increase parts’ homogeneity and process reproducibility. The use of 3.5 bar packing pressure during the polymerization stage resulted in mechanical properties with lower standard deviations, indicating a higher degree of homogeneity of the manufactured parts and higher process reproducibility. These results will be used for establishing the actual state of the technology and will be a base for future process optimization.

## 1. Introduction

Climate changes are one of the main adversities of the XXI century with serious consequences for biodiversity preservation and available resources. In 2019, road transport was responsible for 12% of greenhouse gas emissions from fuel combustion. Aware of this situation, in the last years, the automotive industry has been developing new solutions for the reduction of gaseous emissions, driven by strict regulations and directives issued by countries or international organizations [1,2].

Lightweight design is a strategy followed by the automotive industry to reduce CO_2_ emissions on road transport. With the ongoing transition towards electric mobility, weight reduction can also contribute to enhancing the range of an electric vehicle. One of the heaviest parts of an automobile is typically its body-in-white (BiW) structure. Steel has been the most common material used on automotive structural parts due to its mechanical performance; however, it contributes significantly to the final weight of the car. The weight percentage of the BiW structure in a conventional car is typically around 20%; thus, it is essential to develop a new generation of lightweight materials and processing techniques, ensuring the required mechanical behavior [3,4,5,6,7,8].

Polymer-based materials have been increasingly used by the automotive industry due to their low density and high corrosion and fatigue resistance. To meet the requirements for automotive structural applications, glass and carbon fiber reinforcements have been applied. Thermosetting composites are used in BiW, although these composites are often considered non-recyclable especially when compared to thermoplastics and metals. New technologies and more sustainable materials are necessary to achieve weight reduction in the automotive sector [9,10].

Thermoplastic composites can be manufactured by a liquid molding technology named Thermoplastic Resin Transfer Molding (T-RTM). Through this technology, it is possible to produce polyamide 6 (PA6), polyamide 12 (PA12), poly(butylene terephthalate) (PBT), poly(methyl methacrylate) (PMMA), poly(ε-caprolactone) (PCL) and poly(L-actide) (PLA) [11,12]. Due to its low melt viscosity, commercial availability and low processing temperature, ε-caprolactam is one of the most promising thermoplastic monomers for woven and non-woven fibers’ impregnation. In the presence of a catalyst and an activator, anionic ring-opening polymerization (AROP) of ε-caprolactam monomer occurs inside the mold, resulting in the production of anionic PA6 [13,14].

The industrialization of T-RTM technology faces several technical-scientific challenges, which are mainly due to the lack of process reproducibility and therefore parts’ inhomogeneity, which can be assessed by differences in the mechanical performance within a part and between different parts [13]. Some strategies that have been investigated for improving the T-RTM process focused on the highly reactive nature of the resins and their sensibility to moisture and oxygen content, which can affect the polymerization and quality of the materials produced. Wendel et al. [15,16,17] studied the influence of moisture on APA6 properties. By adding water to the reactive system, the authors observed that moisture hinders the polymerization reaction, causing a decrease in the monomer conversion degree, maximum tensile strength and Young’s modulus of the produced parts. For this reason, it is essential to have a humidity-controlled environment during the process.

Other authors have studied the effect of mold temperature on the properties of the manufactured parts [18,19,20,21,22]. Van Rijswick et al. [18] and Teuwen et al. [22] obtained higher mechanical properties at a polymerization temperature between 150 and 160 °C, showing that the degree of crystallinity and the monomer conversion degree is influenced by the mold temperature. Dong et al. [21] studied the effect of thermal gradients between the resin and the mold in the homogeneity of the composite parts. Analyzing the mechanical properties, a higher homogeneity was achieved when the initial mold temperature was set at the resin injection temperature (100 °C) and after injection heated up to a polymerization temperature (140–170 °C). Higher mechanical properties, but a lower homogeneity, are obtained when the mold is already at the polymerization temperature at the beginning of the injection process. An unbalanced polymerization and crystallization throughout the part seem to occur when the mold is set at these temperature conditions.

The occurrence of resin shrinkage during polymerization can also influence the density, ntnt and geometry of the parts [3,18,23]. Van Rijswijk et al. [18] indicated that, in this process, a typical 9% shrinkage can lead to void formation. Shrinkage should be compensated by providing additional low-viscosity resin into the mold cavity. Teuwen et al. [23] analyzed the effect of the filling pressure, after infusion, in a vacuum-assisted T-RTM process. The authors concluded that a packing pressure applied during the polymerization process can be used to decrease the void volume content (VVC) in composite parts.

By the analysis of the state-of-the-art, it can be inferred that the studies addressing the robustness of the T-RTM process towards its application in an industrial context are still incipient. For establishing a reliable process, it is important to understand the effect of processing parameters in the final part properties. This research analyzes the influence of packing pressure, throughout the filling and polymerization stages, on the homogeneity of the parts and the process reproducibility. The effect of this process parameter was evaluated through physical, thermal and mechanical analysis.

## 2. Materials and Methods

### 2.1. Materials

For this work, AP-Nylon^®^ caprolactam monomer (CL), Brüggolen^®^ C1 catalyst (C1) and Brüggolen^®^ C20P activator (C20P) were used in an 85:10:5 wt% proportion (L. Brüggemann GmbH & Co. KG, Heilbronn, Germany). The chemical structures of the materials used in this work are presented in Figure 1.

Four plies of Saertex X-E-573 g/m^2^–1270 mm biaxial (−45/+45°) non-crimp glass fibers (GF) fabric (SAERTEX GmbH & Co. KG, Saerbeck, Germany) were used as reinforcements.

### 2.2. T-RTM Prototype Equipment and Processing

Neat PA6 and composite parts were produced in T-RTM prototype lab equipment (ESAN, University of Aveiro), according to Figure 2. The equipment was placed in a controlled lab environment with a temperature of 26 ± 3 °C and a relative humidity of ≤45%.

The first processual methodology stage was the inertization of the equipment with nitrogen gas. The CL, C1 and C20P were inserted into a container and heated for 15 min at 95 °C. The raw materials mixture was made by mechanical stirring at 250 rpm for 10 min. Before the injection phase, the molten material was transferred through a polyvinyl chloride inlet hose to the mold cavity.

A single cavity mold was designed to produce plate-shaped parts with a 120 × 75 × 1.8 mm^3^ cavity (Figure 3). The cavity had a U-shape geometry for woven fiber clamping. The mold was heated using electrical cartridge resistors. Due to the low resin viscosity and to allow a vacuum, the mold was sealed using double O-rings. For the composites’ production, 4 GF layers with −45°/+45° layout were placed inside of the mold cavity for approximately 50% of fiber volume.

To fill the resin into the mold, the injection stage occurred under a pressure of 2 bar using nitrogen gas for 30 s. The mold cavity pressure was set at 0.15 bar, and the setpoint temperature of the mold was 160 °C. Preliminary injection tests were performed to set suitable ranges for the injection pressures.

The polymerization time was 30 min for all parts. During polymerization, one or two packing pressures were applied. The first packing pressure stage was set at an inlet mold pressure of 3 bar for two minutes. An optional second stage was implemented to achieve a more efficient control of material backflow, shrinkage and void formation. This stage consisted of the application of 3.5 or 6.0 bar in the mold outlet until the end of the polymerization time. The produced part was then demolded.

Table 1 displays the nomenclature assigned to the parts based on the six processing conditions used. For each manufacturing condition, five parts were produced.

### 2.3. Materials Characterization

The mechanical behavior, monomer conversion degree, density, fiber volume percentage and void content were assessed from different areas of each part, according to Figure 4.

A representative image of the neat polymer and composite specimens, obtained after cutting the part, is depicted in Figure 5.

#### 2.3.1. Mechanical Analysis

Tensile tests were carried out on a Shimadzu Autograph AG-IS 10 kN universal testing machine (Shimadzu Corporation, Kyoto, Japan) at ambient temperature. Tensile tests applied to neat polymers were performed according to ISO 527-2 standard (Type 1BA) at a constant speed of 1 mm/min. Composite specimens tests were performed according to ISO 527-4 standard and conducted at a constant speed of 2 mm/min, with dimensions of 70 × 7 × 1.8 mm^3^ and a gauge length of 14 mm.

Young’s modulus was calculated using a video extensometer in order to measure the gauge length elongation of the specimens.

Figure 6 summarizes the method for assessing the reproducibility and homogeneity of the parts through the studied properties: yield strength (σ_y_), maximum tensile strength (σ_M_) and Young’s modulus.

Average standard deviations of the studied properties measured in all areas (sA), used to evaluate the part-to-part reproducibility [24] of the process, were calculated using Equations (1) and (2) [25].
(1)$$\mathrm{sA}\;\left(\%\right)=\frac{\sum_{\mathrm{Ai}=1}^{\mathrm{NA}}({\mathrm{sA}}_{i}\;\left(\%\right))}{\mathrm{NA}},$$
in which
(2)$${\mathrm{sA}}_{i}\left(\%\right)=\frac{\sqrt{\frac{\sum_{\mathrm{Pi}=1}^{\mathrm{NP}}{\left({\mathrm{Part}}_{\mathrm{Pi}}-u_{\mathrm{Ai}}\right)}^{2}}{\mathrm{NP}-1}}\;}{u_{\mathrm{Ai}}}\times 100,$$
where A_i_ is the area number, NA is the total number of areas, P_i_ is the part number, Part_Pi_ is the studied properties of part Pi (in area Ai), sA_i_ is the standard deviation of studied properties measured in area Ai, NP is the total number of parts and u_Ai_ is the average studied properties measured in area Ai.

Average standard deviations of the properties measured in all plates (sP), used to evaluate the parts homogeneity, were calculated using Equations (3) and (4) [25].
(3)$$\mathrm{sP}\;\left(\%\right)=\frac{\sum_{\mathrm{pi}=1}^{\mathrm{NP}}({\mathrm{sP}}_{i}\;\left(\%\right))}{\mathrm{NP}},$$
in which
(4)$${\mathrm{sP}}_{i}\left(\%\right)=\frac{\sqrt{\frac{\sum_{\mathrm{Ai}=1}^{\mathrm{NA}}{\left({\mathrm{area}}_{\mathrm{Ai}}-u_{\mathrm{Pi}}\right)}^{2}}{\mathrm{NA}-1}}\;}{u_{\mathrm{Pi}}}\times 100,$$
where sP_i_ is the standard deviation of the property measured in part P, area_Ai_ is the property measured in area Ai (in part Pi) and u_Pi_ is the average property measured in part Pi.

#### 2.3.2. Monomer Conversion Degree

The monomer conversion degree was obtained from thermogravimetric analysis (TGA). The tests were conducted on Hitachi STA300 equipment (Hitachi, Ltd, Ibaraki, Japan) at 10 °C/min heating rate, between 25 °C and 550 °C, in open aluminium pans. An inert atmosphere was set by a 200 mL/min nitrogen gas flow. The conversion degree was calculated according to Equation (5) [26].
(5)$$\mathrm{Monomer}\;\mathrm{conversion}\;\mathrm{degree}\;\left(\%\right)=\left[1-\left(\frac{{\mathrm{wl}}_{240\;℃}-{\;\mathrm{wl}}_{100\;℃}}{w_{\mathrm{initial}}-{\;\mathrm{wl}}_{100\;℃}}\right)\right]\times 100,$$
where wl_100 °C_ is the sample weight loss at 100 °C, wl_240 °C_ is the sample weight loss at 240 °C and w_initial_ is the sample initial weight.

#### 2.3.3. Density, Fiber Volume Content and Void Volume Content

The density of the specimens was obtained based on the ISO 1183 standard, by immersion method in water, through an Ohaus Adventurer AX224 analytic balance (Ohaus Corporation, Parsippany, NJ, USA), with its respective density determination kit. In neat polymer parts, the density was measured in the inlet, central and outlet areas (Figure 3). In composite parts, the density was analyzed in the inlet area. For each analyzed area, the densities of five samples were measured.

The fiber volume and void content of the composites were determined based on ISO 7822:1999, Method A standard, known as the burn-off technique. The samples were placed in a Termolab MLM furnace (Termolab, Águeda, Portugal) at room temperature, heated at 10 °C/min until 560 °C and kept at 560 °C for 2 h to eliminate the organic phase. Then, the samples were cooled to room temperature and weighed to determine the fiber weight content (FWC) by the ratio between final weight and initial weight. Fiber volume content (FVC) was calculated according to Equation (6) [3,27].
(6)$$\mathrm{FVC}\;\left(\%\right)=\frac{\mathrm{FWC}\;\times {\;\rho }_{\mathrm{measured}}}{\rho_{\mathrm{GF}}}\times 100,$$
where ρ_measured_ is the composite density and ρ_GF_ is the density of GF.

The VVC for composite parts was determined by Equation (7) [3,27].
(7)$$\mathrm{VVC}\;\left(\%\right)=\left(\frac{\rho_{\mathrm{theoretical}}-{\;\rho }_{\mathrm{measured}}}{\rho_{\mathrm{theoretical}}}\right)\times 100,$$
where the ρ_theoretical_ is the density estimated considering the FVC and the resin volume content.

Composites VVC and morphology were evaluated from polished samples (finished with 9 µm diamond paste) through a Polarized Optical Microscope (POM) and a Scanning Electron Microscope (SEM). The samples were collected from the inlet area of the plates. A reflected-light Nikon Eclipse L150 (Nikon Corporation, Tokyo, Japan) and a Canon EOS 100D (Canon Inc. Tokyo, Japan) digital single lens reflex camera were used for optical micrographs. The SEM micrographs were obtained using a Tescan Vega LMS (Tescan Orsay Holding, a.s., Brno, Czech Republic) with an accelerating voltage of 30 kV. The samples for SEM were sputter coated during 300 s with a layer of gold-palladium. The images were treated to highlight the presence of voids using ImageJ, version 1.51j8.

## 3. Results and Discussion

The mechanical behavior, monomer conversion degree, density and fibers volume content of the neat polymer and composite parts are going to be evaluated in this paper section.

### 3.1. Mechanical Analysis

PA6 properties depend on its formulation and processing conditions. PA6 yield stress, for instance, can typically range from around 50 to 75 MPa [18,28]. There is usually some variation in the data collected from mechanical tests due to several factors such as the apparatus resolution and calibration and operator measuring errors and some inhomogeneities always exist, even within the same lot of material.

As already mentioned, besides the production of parts under the same processing conditions, to assess process reproducibility, this work also studied the differences in the mechanical performance within each part to evaluate the parts’ homogeneity. It is intended to optimize the second packing pressure variable also based on this evaluation for neat polymer and composite parts [29].

#### 3.1.1. Neat Polymer Parts

The σ_y_, σ_M_ and Young’s modulus for neat polymer parts obtained with different packing pressures are presented in Figure 6. The APA6_3.5 parts achieved higher σ_y_, σ_M_ and Young’s modulus when compared to the parts manufactured without the second packing pressure. The increasing of the packing pressure to 6 bar did not contribute to a further increase in the mechanical properties. This behavior can be due to the nitrogen pressure inducing voids in the resin. The APA6_3.5 parts mechanical behavior is on par with the best properties obtained by AROP of CL through T-RTM [18,30].

In Figure 7 and Figure 8, the average values of sP and sA for each processing condition are also visible. Overall, when the second packing pressure is applied, the values of standard deviations tended to decrease. The application of packing pressure throughout the polymerization stage can compact the resin and compensate for its shrinkage.

Analyzing each part individually, the values of sP tended to be lower for a packing pressure of 3.5 bar, which can indicate a higher parts’ homogeneity. In the specific case of Young’s modulus, a similar sP was found.

In the evaluation of the neat polymer results between different parts, the sA tended to be lower for 6 bar. Admitting that the reproducibility of the process can be measured by the sA, this result can be a sign of a more reliable process.

#### 3.1.2. Composite Parts

The σ_M_ and Young’s modulus of the composites obtained at different packing pressures are shown in Figure 9. Compared to neat polymer parts, composite specimens achieved higher σ_M_ and Young’s modulus due to the presence of fibers, denoting a transfer of properties from the fibers to the composite. As with neat polymers, the parts produced with 3.5 bar achieved higher σ_M_ when compared to the parts without second packing pressure. However, the differences were lower, probably due to the pressure drop caused by the fibers. The increase of the packing pressure to 6 bar also did not lead to a further increase in σ_M_. This behavior can arise from the possibility of void formation due to nitrogen pressure. Although Young’s modulus tended to decrease with the increase of second packing pressure, it can be considered that those differences did not have a significant impact on composite parts’ behavior. The results are within or slightly above the range of values reported in the literature for GF composites obtained by T-RTM [30,31].

Composite sP and sA values for each processing condition are presented in Figure 9 and Figure 10. APA6/GF_3.5 parts had the lowest sP and sA values for σ_M_ and Young’s modulus, which can indicate an improvement in parts’ homogeneity and process reproducibility. An intermediate 3.5 bar packing pressure also led to the lowest sP values in the composite parts.

Figure 11 displays a comparison of the standard deviations associated with the reproducibility (sA) and homogeneity (sP) between neat polymers and composite parts for σ_M_ and Young’s modulus. The results indicate that the parts manufactured with a 3.5 bar second packing pressure promote lower standard deviations. The composite parts tended to have higher standard deviations, which may be explained by a less effective pressure transmission due to the presence of fibers, during injection/packing stages, and also by differences in the permeability of the fibers.

### 3.2. Monomer Conversion Degree

According to the literature, it is important to achieve a conversion degree above 95% to avoid fiber-matrix interface lack of adhesion issues [18,30,32].

All the neat polymer and composite samples analyzed had a conversion degree above 95%. Since oxygen and humidity inhibit the polymerization of raw materials, the equipment was previously pressurized with nitrogen gas. It was also essential to perform the trials in a laboratory with a controlled temperature and humidity environment. A relative air humidity above 45% can promote moisture absorption of the raw materials when they are transferred to the container. Humidity affects resin polymerization through the deactivation of the catalytic system (C1), which occurs due to its reaction with water, leading to the formation of secondary products. Since the polymerization kinetics is influenced by temperature, the environment’s thermal stability is also important for the manufacturing process [16].

#### 3.2.1. Neat Polymer Parts’ Monomer Conversion Degree

The neat polymer monomer conversion degree is displayed in Figure 12. The conversion degree was around 98% with standard deviation values below 1% for all the experimental conditions. The reduced differences in conversion degree between the inlet, center and outlet areas and between distinct parts were an indication of suitable parts’ homogeneity and process reproducibility. The monomer conversion degree for neat polymer parts was within the range of the studies for AROP of CL by T-RTM [8,9,13].

#### 3.2.2. Composite Parts’ Monomer Conversion Degree

The composites monomer conversion degree, by part and part area, is shown in Figure 13. For the composite parts, the conversion degree was ~99%, a value slightly higher than that observed in neat polymer. This can be explained by higher thermal stability in the mold cavity promoted by the presence of the reinforcing fibers. In the presence of fibers, less resin is injected into the mold cavity, decreasing the resin’s thermal inertia during polymerization. In addition, a higher thermal conductivity of GF, compared to the resin, enables a fast and more balanced heat transfer within the mold cavity. The conversion degree values for the composite parts were similar to the values reported in the literature [22,30,31,33].

### 3.3. Density, FVC and VVC

#### 3.3.1. Neat Polymer Parts’ Density and VVC

The densities of the neat polymer by part and part area are presented in Figure 14. The results suggest that the increase of the second packing pressure can have a negative effect on part densities. This is in agreement with the possibility that nitrogen gas pressure can induce voids in the resin [18]. Although the mentioned density differences are not very substantial, even a small variation in neat polymer densities can lead to a significant increase in VVC. Since this effect is enhanced by increasing nitrogen pressure, this could explain the lower mechanical behavior of APA6_6.0 when compared to APA6_3.5 parts. The density standard deviation values were below 1% for all the processing conditions.

Considering an APA6 theoretical density of 1.16 g/cm^3^ (for a 100% dense APA6 part) [18,32,34], a lower density can be attributed to the presence of 1 to 1.5% VVC in neat polymer samples [3,35].

#### 3.3.2. Composite Parts’ Density, FVC and VVC

The densities of the composites by part are shown in Figure 15. Compared to neat APA6, the densities of the composites are higher due to the presence of the fibers. The average densities of the composite parts were similar.

Standard deviation values are ranging from 1 to 2%. The higher standard deviation on the density of the composite parts can be attributed to the differences in the fiber volume content (Figure 15a). An increase in fiber volume content tended to lead to a higher part density. The obtained fiber volume content of 55–60% is higher than what is usually presented in the literature for this technology [22,23,27,30,31].

Young’s modulus was also affected by the differences in fiber volume content (Figure 15b). For each packing condition, it can be observed that an increase in fiber volume content lead to an increase in Young’s modulus [36]. This can explain the higher standard deviation values in composite Young’s modulus results.

The burn-off technique was used for composite samples to evaluate the influence of the packing pressure on VVC. The results are summarized in Table 2 and indicate that the increase of the packing pressure can have a negative effect on VVC.

The burn-off technique is known for its low accuracy, which can be particularly noticeable in samples with a VVC up to 2%. The low accuracy is reflected in the standard deviation values presented. Therefore, POM images were used to evaluate composites’ VVC and morphology. Representative POM images for each processing condition are depicted in Figure 16a–c, in which it is possible to see the GF layers as horizontal lines. In samples obtained using 6 bar of packing pressure, it was possible to identify more voids, namely, macro voids, (Figure 16a–c). In order to highlight the voids’ presence, treated images (imageJ) are presented in Figure 16d–f.

SEM micrographs showed a similar tendency verified in POM analysis (Figure 17a–c). As expected, it was possible to identify more voids in the APA6/GF_6.0 samples, including macro voids, as can be seen in the edited images in Figure 17d–f. This is an indication that nitrogen presence induces void formation. Such presence compromises the composite mechanical behavior.

## 4. Conclusions and Future Work

The use of a 3.5 bar second packing pressure had a positive influence on APA6 tensile mechanical behavior. An increase of the packing pressure to 6 bar did not contribute to a further increase in the tensile properties, due to the presence of a higher VVC induced by nitrogen gas.

The results suggest that the packing pressure can aid to increase process reproducibility in the production of neat polymer parts, particularly when using 6 bar. An intermediate 3.5 bar packing pressure can also contribute to increasing APA6 parts’ homogeneity.

In composite parts, packing pressures did not contribute to a significant increase in mechanical properties. This could be due to the pressure barrier created by the fiber and to the increase of the VVC for higher packing pressure.

For composite manufacturing, the results indicated that the use of a 3.5 bar intermediate packing pressure can contribute to increasing the process reproducibility and the homogeneity of the parts.

A homogeneous and reproducible monomer conversion was obtained for neat polymer and for composite parts indicating that the experimental procedure was reliable.

Polymer and composite parts’ densities suggest that the increase of the second packing pressure can have a negative effect on parts due to nitrogen-induced voids.

In future work, to improve process reproducibility and parts’ homogeneity, an alternative pressure medium should be evaluated to avoid nitrogen-induced voids. To improve the mechanical properties of composites, a higher packing pressure should be addressed.

In situ dielectric monitoring can also be a particularly useful tool to understand the phenomena involved during polymerization and to assess the reaction reproducibility and quality control in real-time.

## Figures and Tables

**Figure 1 materials-16-04652-f001:**
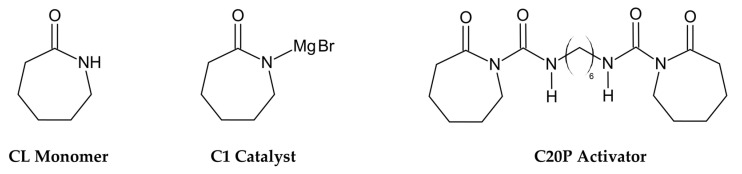
Chemical structures of the materials used.

**Figure 2 materials-16-04652-f002:**
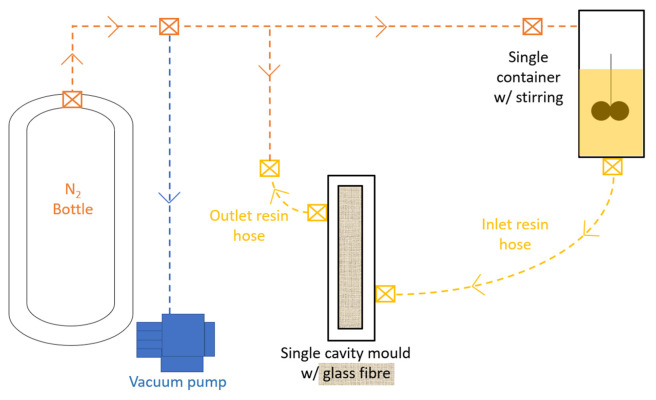
Scheme of the prototype T-RTM equipment.

**Figure 3 materials-16-04652-f003:**
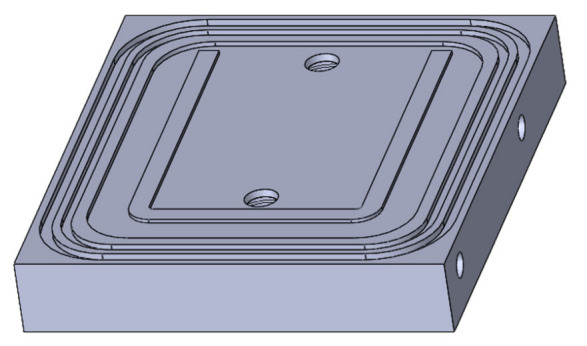
CAD 3D image of the mold cavity with the U-shape geometry.

**Figure 4 materials-16-04652-f004:**
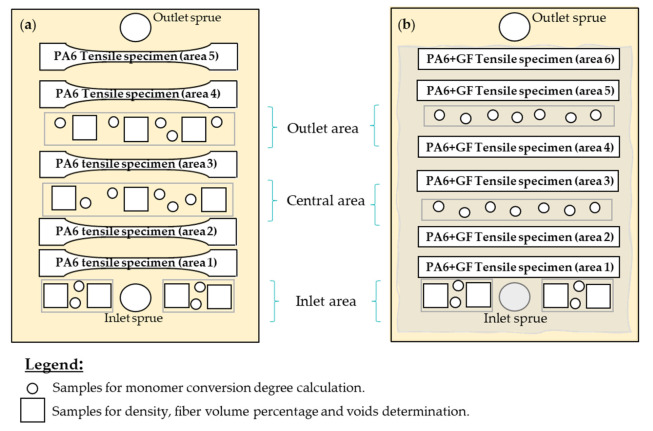
Localization of samples on (**a**) neat polymers and (**b**) composite parts.

**Figure 5 materials-16-04652-f005:**
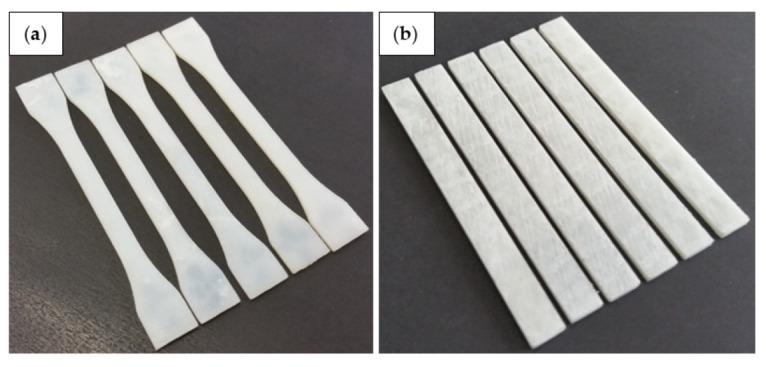
Examples of (**a**) APA6_3.5 and (**b**) APA6/GF_3.5 specimens after cutting.

**Figure 6 materials-16-04652-f006:**
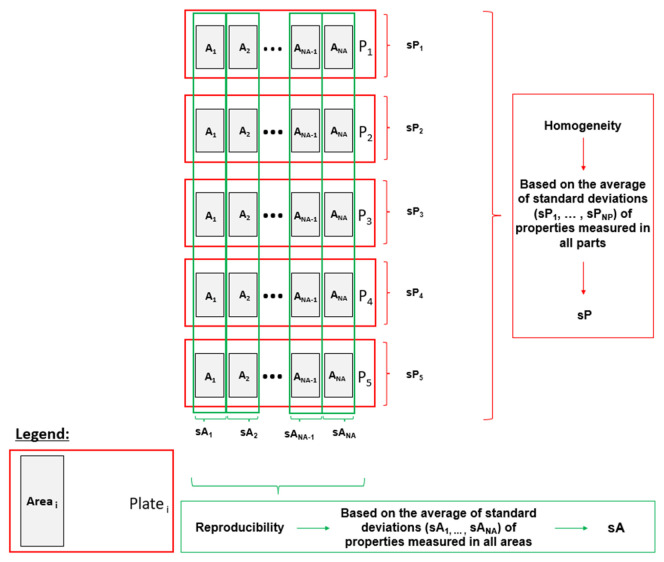
Scheme of the methodology used to assess the homogeneity and reproducibility of the parts.

**Figure 7 materials-16-04652-f007:**
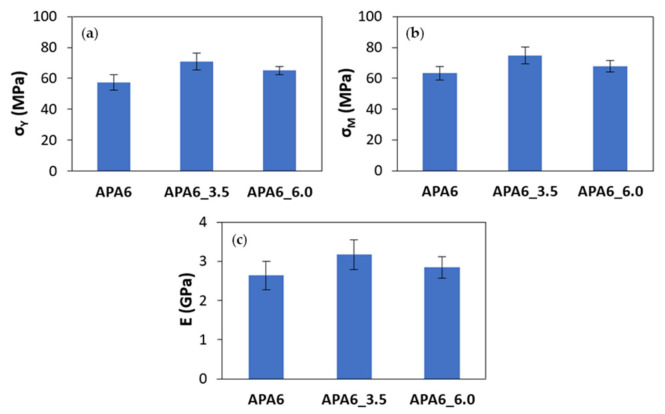
Mechanical properties of neat polymer parts: (**a**) yield stress (σ_y_), (**b**) maximum tensile strength (σ_M_) and (**c**) Young’s modulus (E).

**Figure 8 materials-16-04652-f008:**
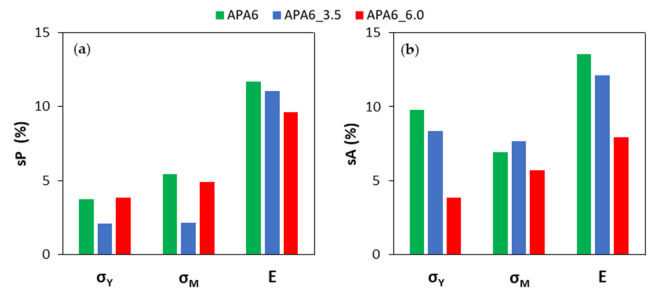
Standard deviations for neat polymers’ mechanical properties (**a**) in each part (sP) and (**b**) from part to part (sA).

**Figure 9 materials-16-04652-f009:**
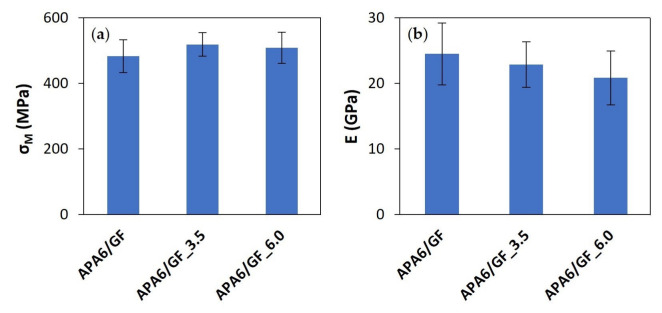
Mechanical properties of composite parts: (**a**) maximum tensile strength (σ_M_) and (**b**) Young’s modulus (E).

**Figure 10 materials-16-04652-f010:**
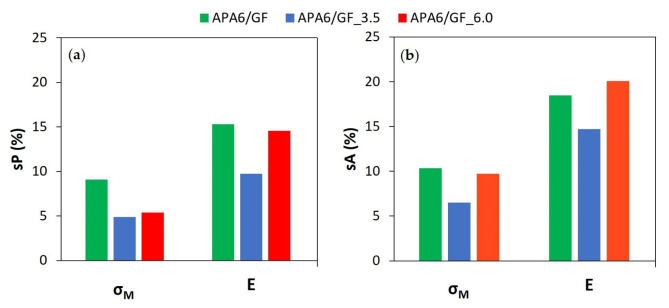
Standard deviations for composites’ mechanical properties: (**a**) in each part (sP) and (**b**) from part to part (sA).

**Figure 11 materials-16-04652-f011:**
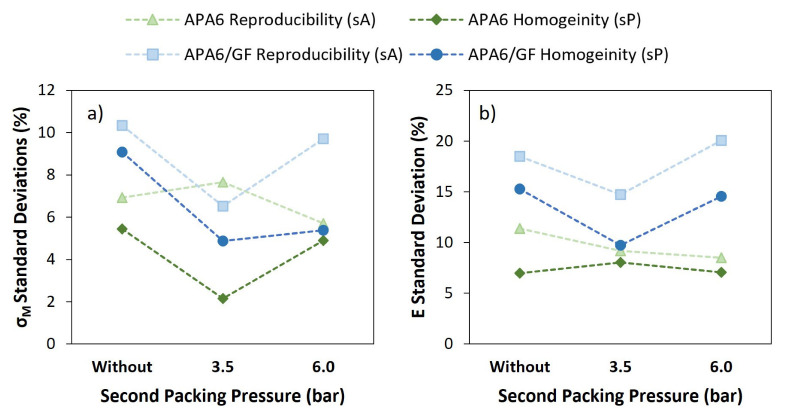
Standard deviations for neat polymers’ and composites’ mechanical properties: (**a**) maximum tensile strength (σ_M_) and (**b**) Young’s modulus (E).

**Figure 12 materials-16-04652-f012:**
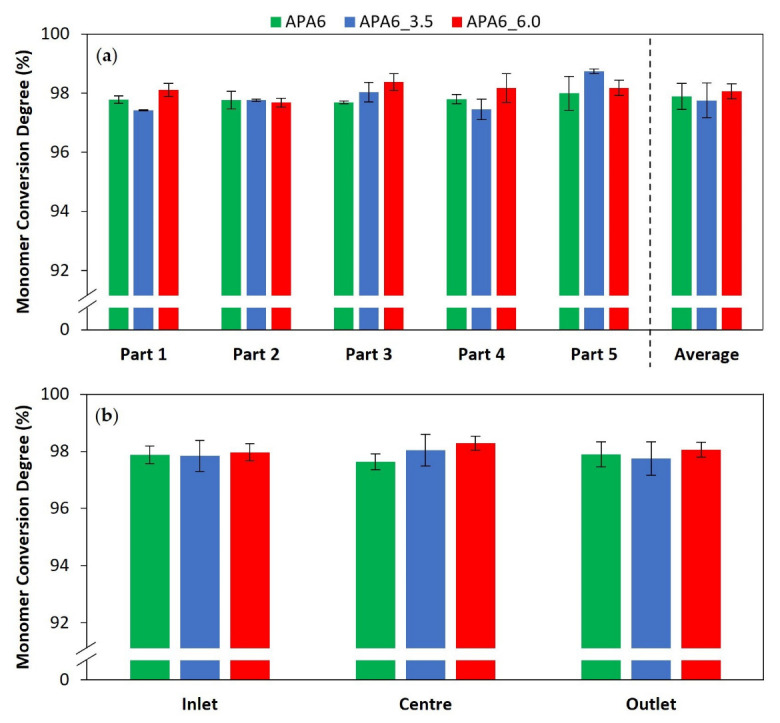
Monomer conversion degree of neat polymer parts (**a**) by part and (**b**) part area.

**Figure 13 materials-16-04652-f013:**
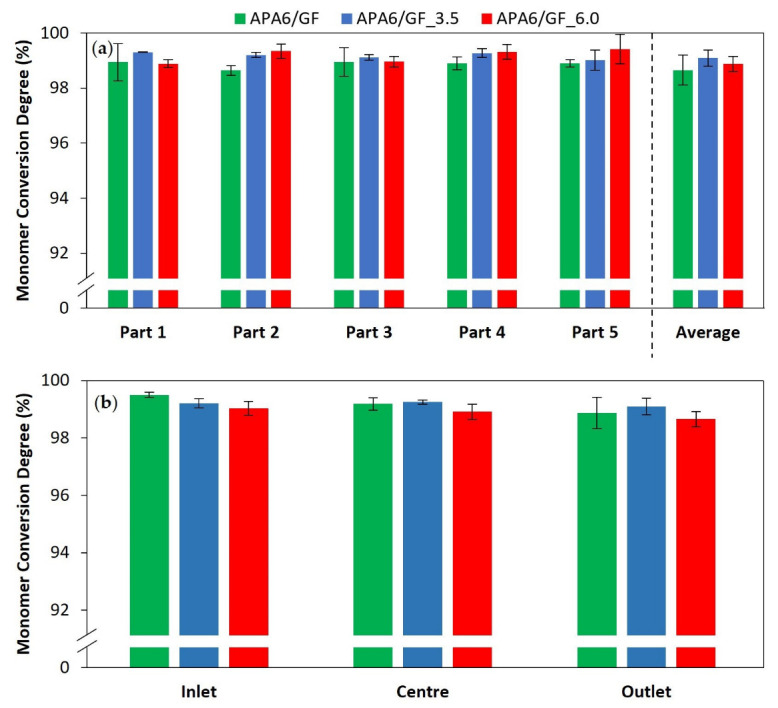
Monomer conversion degree of composite parts (**a**) by part and (**b**) part area.

**Figure 14 materials-16-04652-f014:**
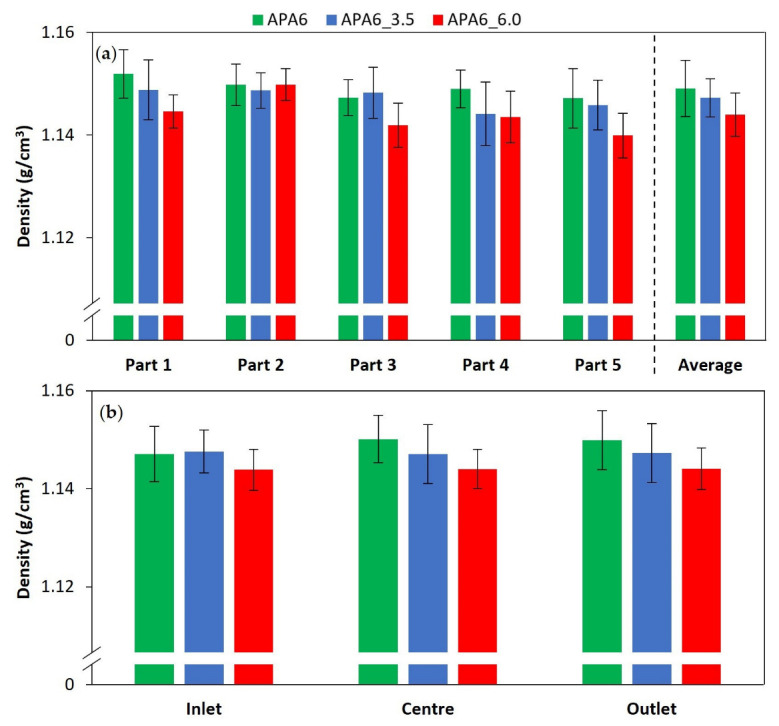
Density of composite parts (**a**) by part and (**b**) part area.

**Figure 15 materials-16-04652-f015:**
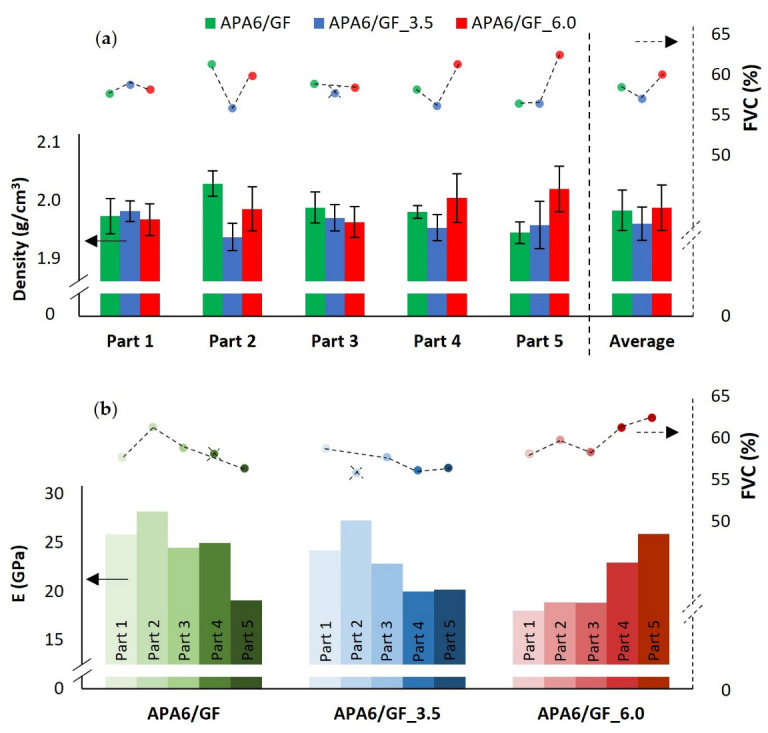
Composite parts’ (**a**) density and (**b**) Young’s modulus (E) compared to fiber volume content (FVC) percentage.

**Figure 16 materials-16-04652-f016:**
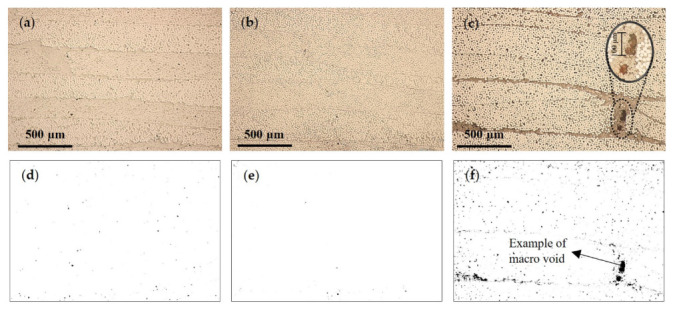
Composite POM images of (**a**) APA6/GF, (**b**) APA6/GF_3.5, (**c**) APA6/GF_6.0, (**d**) APA6/GF voids, (**e**) APA6/GF_3.5 voids and (**f**) APA6/GF_6.0 voids.

**Figure 17 materials-16-04652-f017:**
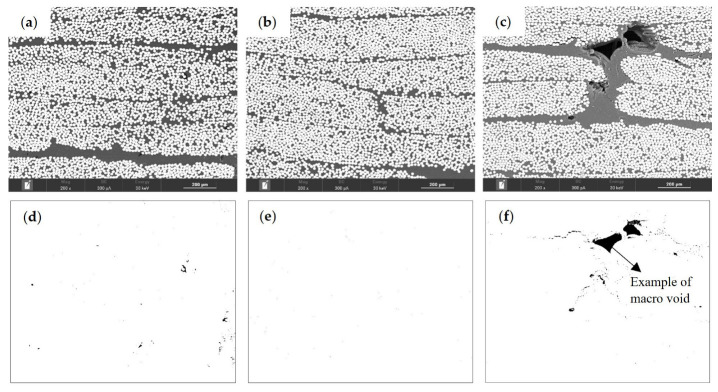
Composite SEM micrographs of (**a**) APA6/GF, (**b**) APA6/GF_3.5, (**c**) APA6/GF_6.0, (**d**) APA6/GF voids, (**e**) APA6/GF_3.5 voids and (**f**) APA6/GF_6.0 voids.

**Table 1 materials-16-04652-t001:** Parts’ nomenclature accordingly to reinforcements and packing pressure conditions.

Nomenclature	Reinforcement Phase	First Packing Pressure (bar)	Second Packing Pressure (bar)
APA6	-	3.0	-
APA6/GF	GF	3.0	-
APA6_3.5	-	3.0	3.5
APA6/GF_3.5	GF	3.0	3.5
APA6_6.0	-	3.0	6.0
APA6/GF_6.0	GF	3.0	6.0

**Table 2 materials-16-04652-t002:** Composites’ VVC obtained by burn-off technique.

Parts	VVC (%)
APA6/GF	1.0 ± 0.6
APA6/GF_3.5	1.0 ± 0.5
APA6/GF_6.0	1.8 ± 0.3

## Data Availability

Data is contained within the article.

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
