# Peer review of "Reproducibility Study of the Thermoplastic Resin Transfer Molding Process for Glass Fiber Reinforced Polyamide 6 Compositesâ€"

_materials, 2023, doi:10.3390/ma16134652_

Round 1
Reviewer 1 Report
Dear authors,
it is a very good paper. So I only have a few comments.
1) The second series (d-f) of the microscopy images are very good results, but with 6e, for example, you can't see anything at all. Perhaps the second series could be completely integrated into the first series, as was done with 6c and 6f.
2) I found some spelling mistakes also in the heading:
3.2 Monomer conversion degree
3.1. Mechanical analisis
3) and a few more formal errors:
Table 1 is duplicate
many: errors! Reference source not found
Both depends on the software.
Please check the final version on 2) and 3) before publishing
BR
The reviewer
Author Response
Response 1: Compared to “f” images, “d”/”e” images have fewer highlighted voids, which makes them more difficult to visualize. Considering those differences representative, based on the set of images analyzed, we tried to show the VVC differences observed.
Response 2: The suggestion was accepted and corrected by the authors.
Response 3: The suggestions were corrected by the authors.

Reviewer 2 Report
This manuscript studies the properties of polyamide 6 reinforced by glass fiber in thermoplastic resin transfer molding process. The manuscript has potentials; however, it needs some revisions before the final decision.
1. Please quantitively present some results in the abstract.
2. One of the distinguishing features of academic writing is that it is informed by what is already known, what work has been done before, and/or what ideas and models have already been developed. As well as being systematic, the review should be evaluative and critical of the studies or ideas which are relevant to the current work. The Introduction does not have sufficient literature survey and it should be significantly improved.
3. The novelties and contributions of the study should be clearly presented in a separate paragraph of the Introduction. It is not clear what are the novelties of the study. These questions should be clearly answered in this paragraph: What are the main questions addressed by the research? Does it address a specific gap in the field and what is this gap? What does it add to the subject area compared with other published material?
4. Line 88 “95°C”, please avoid missing spaces between values and units. Each value should have a space with unit such as “95 °C”. Please address this issue throughout the manuscript.
5. Equations (3) and (4) are almost the same with Equations (1) and (2) and only the parameter is changed. It is recommended to present these equations in general form and eliminate two of them.
6. It is necessary to provide supporting references for all equations.
7. Please resolve the caption errors, for instance, in Line 202 and Line 209.
8. For a better comparison, it is necessary to compare the results of APA6 and APA6/GF in one figure.
9. What is the reason of eliminating the yield stress from Figure 8?
10. The standard deviations of the void volume content results in Table 2 are very large. This affects the accuracy of the measurements.
Author Response
Response 1: We considered that the most important results were based on the observation of tendencies rather than on quantitative values. In line with the suggestion made, text was added to the abstract (please see the uploaded documents).
Response 2: The suggestion was accepted and corrected by the authors. Please consider the information added to the introduction.
Response 3: The suggestion was accepted by the authors. The text was added to the introduction.
Response 4: The suggestion was accepted and corrected by the authors.
Response 5: We understand the idea, however, the purpose of presenting the equations separately was to make a clear distinction between the equation adopted to assess the homogeneity of the plates and the reproducibility of the process. These equations are also related with Figure 6: “Scheme of the methodology used to assess the homogeneity and reproducibility of the parts”. Our concern is that putting the equations together will make the distinction between the two concepts more difficult to understand and confusing.
Response 6: The suggestion was accepted and introduced by the authors.
Response 7: The suggestion was accepted and corrected by the authors.
Response 8: The suggestion was accepted by the authors. Please see the new Figure 11: “Standard deviations for neat polymers and composite mechanical properties: (a) maximum tensile strength (σM) and (b) Young´s modulus (E).”
Response 9: In APA6/GF composites, there was no yield stress. According to ISO 527-1, the APA6 specimens had a typical stress/strain curve similar to curve b (tough material with yield point), while APA6/GF specimens had a mechanical behavior similar to curve a (brittle material).
Response 10: We acknowledge that the standard deviation values from the burn-off technique are high. This is a common problem mentioned in the literature. However, we believe that it is important to present these results, however, complemented with a qualitative analysis of POM and SEM. These two combined analyses allow us to better understand the effect of the packing pressure on the occurrence of voids in a qualitative perspective.

Reviewer 3 Report
The researchers studied the influence of packing pressure, as process variable, throughout the filling and polymerization stages of PA. To assess the process reproducibility and parts’ homogeneity, physical, thermal and mechanical properties were analyzed in different areas of neat PA6 and composite parts. This study showed that a two-stage packing pressure can be successfully used to increase parts homogeneity and process reproducibility. The use of 3.5 bar packing pressure during the polymerization stage reached the most promising results. The paper could be published after revision.
-Could other monomer of PA used in this technology ?
-Chemical structures of the materials used should be demonstrated.
-The injection stage occurred under a pressure of 2 bar 99 using nitrogen gas for 30 s. How it was decided that this pressure is suitable for injection.
-There are many „ Error! Reference source not found.“ In the text.
-What in molecular weight of the polymers after polymerization ?
-Values of glass transition or melting temperatures should be obtained.
-The same part should be prepared from commercial PA and properties should be compared ? This would demonstrate if the new technology is advantage.
-Is similar technology of polymerization already described in literature for other polymers ?
Author Response
Response 1: Besides ε-CaproLactam (ε-CL) for PA6 production, based on our knowledge, only PA12 can also be produced by the synthesis of ω-LauraLactam (ω-LL). Besides PA, other materials can also be produced.
Response 2: The suggestion was accepted by the authors. A new Figure 1 was added to the article (please check the uploaded documents).
Response 3: Preliminary injection tests were performed to set a suitable range for the injection pressure. Using a mold with a transparent window it was possible to visualize the injection and subsequent impregnation of the reinforcement fibers according to the pressure and injection time used. Text was added to the article.
Response 4: The suggestion was accepted and corrected by the authors.
Response 5: Please see response 6.
Response 6: Due to its relevance, we are currently preparing a second article that will analyze the thermal behavior and the molecular weight of these materials, so we choose not to include these results in this article.
Response 7: The commercial PA6 has a high molten viscosity that is unsuitable for the impregnation of continuous fibers, and thereby, for the production of thermoplastic continuous reinforced composites. In this technology, the main focus was the production of thermoplastic composites with low-viscosity thermoplastic precursors. We did consider comparing it with available commercial composites, however, they are usually processed with a thermosetting matrix. Because of that, we also didn´t compare the APA6 with commercial PA6. We will try to consider this suggestion for future work.
Response 8: Besides PA6 manufacturing by T-RTM, the anionic ring-opening polymerization (AROP) can also be used for polyamide 12 and Poly(ButyleneTeraphthalate) (PBT) production.
Other thermoplastics can be produced by in-situ polymerization, such as, Poly(MethylMethAcrylate) (PMMA), Poly (ε-caprolactone) (PCL) and Poly(L-actide) (PLA).
Response 8: Besides PA6 manufacturing by T-RTM, the anionic ring-opening polymerization (AROP) can also be used for polyamide 12 and Poly(ButyleneTeraphthalate) (PBT) production. Other thermoplastics can be produced by in-situ polymerization, such as, Poly(MethylMethAcrylate) (PMMA), Poly (ε-caprolactone) (PCL) and Poly(L-actide) (PLA). This work focused on the polymerization of ε-CL to produce an APA6 due to its low melt viscosity, its commercial availability (and its respective reagents), low processing temperatures and polymerization rate. Using ε-CL we are closer to enabling the implementation of this technology at an industrial stage. Text was added to the introduction.

Round 2
Reviewer 3 Report
Accept in present form